# Possible Interaction between Dabigatran and Ranolazine in Patients with Renal Failure

**DOI:** 10.3390/medicina56010013

**Published:** 2019-12-29

**Authors:** Gintautas Gumbrevičius, Gytė Damulevičienė, Vaidotas Galaunė, Milda Gumbrevičiūtė

**Affiliations:** 1Division of Clinical Pharmacology, Institute of Physiology and Pharmacology, Medical Academy, Lithuanian University of Health Sciences, 44307 Kaunas, Lithuania; vaidotasgal@gmail.com; 2Department of Internal Medicine, Kaunas Clinical Hospital, 44127 Kaunas, Lithuania; milda0812@gmail.com; 3Department of Geriatrics, Medical Academy, Lithuanian University of Health Sciences, 44307 Kaunas, Lithuania; gytedamu@gmail.com; 4Department of Geriatrics, Kaunas Clinical Hospital, 44127 Kaunas, Lithuania

**Keywords:** anticoagulants, dabigatran, ranolazine, drugs interaction, renal failure

## Abstract

Dabigatran etexilate is a direct oral anticoagulant (thrombin inhibitor) used for the prevention of stroke and systemic thromboembolic events in patients with permanent atrial fibrillation; prevention of venous thromboembolic events and deep veins thrombosis; treatment and prevention of pulmonary embolism. Dabigatran is a relatively new drug, and as a result, its interactions with other medications and their significance are not fully known. A 72 years old male, having a medical history of heart and renal failure, was hospitalized for pneumonia treatment. The patient was taking several drugs, including dabigatran 150 mg twice daily and ranolazine 750 mg twice daily. His creatinine clearance was 45.22 mL/min, International Normalized Ratio (INR)—7.03. Dabigatran was discontinued. After 9 days, INR decreased to 1.33, and after 6 days, creatinine clearance increased to 64.39 mL/min. The patient was taking an adequate dosage of dabigatran, thus dabigatran was thought to be overdosed due to its interaction with ranolazine because dabigatran is a p-glycoprotein substrate, whereas ranolazine is the inhibitor of this transporter. Dabigatran and ranolazine should be used with caution in patients with renal failure. It is recommended to use smaller doses of both medications and observe coagulation parameters if needed.

## 1. Introduction

Dabigatran etexilate is a direct oral anticoagulant (thrombin inhibitor), used for the prevention of stroke and systemic thromboembolic events in patients with non-valvular atrial fibrillation; prevention of venous thromboembolic events in adult patients who have undergone elective total hip replacement surgery or total knee replacement surgery; treatment of deep vein thrombosis or pulmonary embolism and prevention of its recurrence [1]. Bleeding is the most common side effect of dabigatran [1,2,3]. Ranolazine is indicated as an add-on therapy for patients with stable angina who are inadequately controlled or intolerant to first-line anti-anginal therapies [4]. These two medications are sometimes taken together. Dabigatran is a relatively new drug, and as a result, not all of its interactions with other medications and their significance are fully known. We would like to discuss the possible interactions between dabigatran and ranolazine.

## 2. Case Report

A 72-year-old man presented to the emergency department with a 3-day history of cough, fever, fatigue, and dyspnea after medium-intensity physical exercise. He had a medical history of arterial hypertension, myocardial infarction, non-valvular atrial fibrillation, 3rd functional class heart failure (New York Heart Association), type 2 diabetes mellitus, diabetic nephropathy with secondary anemia, and 2nd stage renal failure. 10 years ago he had undergone surgical treatment of prostatic cancer. The drugs he used were dabigatran (Pradaxa) 150 mg twice a day (the patient was taking last doses in the morning for one day before hospitalization), aspirin 100 mg once a day, isosorbide mononitrate 40 mg once a day, ranolazine (Ranexa) 750 mg twice a day, torasemide 20 mg once a day, doxazosin 4 mg once a day, fixed-dose combination of olmesartan 40 mg, amlodipine 10 mg and hydrochlorothiazide 12.5 mg once a day, pre-mixed insulin aspart/protamine-crystallized and insulin aspart in the ratio of 30/70, 3 times per day (34 IU and 24 IU). This study was approved by Kaunas Regional Ethical committee, approval number BE9-12, approved on 12 December 2019. Written informed consent was obtained from the patient involved in this study.

The patient’s weight was 91 kg. Upon physical examination, the patient’s lips were slightly cyanotic, peripheral oxygen saturation was 96% while breathing room air, chest auscultation revealed rales at the base of the left lung, and heart rate was irregular, 76 bpm. Chest radiography indicated an infiltration at the base of the left lung. Laboratory test results showed an elevated C-reactive protein (CRP) 174 mg/L; elevated serum creatinine 168 μmol/L (Table 1), creatinine clearance (CrCl) 45.22 mL/min; microcytic hypochromic anemia (Hb 95 g/L, MCV−83.5 fl), INR 7.03, prothrombin index 6%. ECG showed atrial fibrillation, left bundle branch block, ventricular extrasystoles, and QTc was 473 msec (formula Bazett). No signs of bleeding were detected. The patient scored 24 points in the Mini-Mental State Examination (MMSE) (mild cognitive impairment). The plasma concentration of dabigatran was not evaluated. Based on clinical findings, laboratory tests, and chest radiography, pneumonia of the lower right lobe was diagnosed.

The patient had no symptoms of hepatic failure, and he was not using vitamin K antagonists. The patient had not used one or more extra doses of dabigatran. Based on the fact that the INR value was significantly above the therapeutic range, the administration of dabigatran was discontinued. Other drugs at hospitalization were kept at the same doses. Laboratory tests, repeated one day after the discontinuation of dabigatran, showed a decreased value of INR to 4.95. The value of INR continued to decrease in the upcoming days and 3 days after it reached 2.88, whereas QTc had shortened to 414 msec in ECG. At that point, dabigatran was restarted with a decreased dose of 150 mg once a day. INR continued to decrease during the whole hospitalization period, which lasted for 9 days. The final INR was obtained on day 9, just before discharge, the result was 1.33. Pneumonia was successfully treated with ceftriaxone.

## 3. Discussion

Dabigatran etexilate is a substrate for the efflux transporter P-gp. The bioavailability of dabigatran following oral administration of Pradaxa capsules is approximately 6.5%. Peak plasma concentration (Cmax) of dabigatran is achieved in 1.25–2 h, the elimination half-life is 12–17 h [1,5]. Cmax and the area under the concentration curve (AUC) are dose-proportional. About 80%–85% of dabigatran is primarily excreted by the kidneys. Dose adjustments are necessary in patients with moderate renal impairment (creatinine clearance (CrCl) 30–50 mL/min) because the AUC of dabigatran can become 2.7 times higher. The recommended dose of dabigatran is 300 mg (i.e., 150 mg twice a day) in patients with atrial flutter and moderate renal impairment. Dabigatran dose in patients with moderate renal impairment should be reduced to 220 mg (i.e., 110 mg twice a day) [1,2] if patients have a high bleeding risk. Dabigatran is contraindicated in patients with CrCl lower than 30 mL/min [1].

It is not necessary to observe anticoagulative parameters while using dabigatran. If there are some additional risk factors, measuring dabigatran depended on the anticoagulative effect, which can help to identify overdosing. Evaluating plasma-diluted thrombin time (DTT), ecarin clotting time (ECT), and activated partial thomboplastin time (aPTT) can be helpful [1,2,6,7]. The international normalized ratio (INR) is not reliable and is not recommended to be measured in patients taking dabigatran. Dabigatran increases the INR in a concentration-dependent manner defined by a nonlinear relationship, and in most cases, its increase is not significant [6,7]. INR significantly increases only when the dabigatran plasma concentration is above the therapeutic range [6]. Some cases were reported when overdosing dabigatran caused INR to increase to 2.6 or even to 8.8 [8,9,10]. Usually, it occurred in patients with severe renal impairment [8,9,10].

Elevated patient‘s INR (7.03) leads to suspecting a highly increased dabigatran plasma concentration. Despite the fact that the patient had moderate renal impairment (CrCl 45.22 mL/min), he was taking the standard dose of dabigatran based on the summary of product characteristics (SmPC). Dabigatran couldn‘t be overdosed because of renal impairment. *European Society of Cardiology* (*ESC*) recommends reducing dabigatran dose only in patients with CrCl 30–49 mL/min and higher bleeding risk [2]. This patient did not have any hepatic disease or clinical signs of hepatic impairment. Moreover, he did not take one or more extra doses of dabigatran.

Dabigatran can also be overdosed due to a drug interaction. There is a confirmed pharmacodynamic interaction between dabigatran and aspirin, but it does not change the INR value [1,2]. In this case, INR decreased from 7.03 to 1.33 in 9 days. Due to this pharmacodynamic interaction, bleeding risk was very high overall these days. In case of a very high bleeding risk, hemodialysis can be performed to shorten this period. One hemodialysis can remove 50%–60% of dabigatran [1]. Dabigatran is not metabolized by Cytochrome CYP450 enzymes but is the substrate of P-gp [1,2]. Coadministration of dabigatran with P-gp inhibitors may significantly increase its bioavailability and plasma concentration. For example, taking dabigatran with ketoconazole AUC_0-∞_ and C_max_ increases, respectively, 2.53 times and 2.49 times; taking with dronedarone—2.4 and 2.3 times; taking with amiodarone—1.6 and 1.5 times [1,2]. A case was reported, where the patient with renal failure was taking dabigatran together with P-gp inhibitor amiodarone. As a result, plasma dabigatran concentration increased 25 times [11]. In patients with moderate renal impairment and concomitantly treated with another P-gp inhibitor verapamil, AUC of dabigatran can increase significantly, and a dose reduction of dabigatran to 75 mg daily should be considered [1]. In this case, only one drug used by the patient could possibly block P-gp and it is ranolazine [4,12]. Ranolazine may have antianginal effects by inhibition of the late sodium current in cardiac cells [4,13]. The bioavailability of ranolazine after oral administration varies from 35% to 50%. The majority of absorbed ranolazine undergoes rapid and extensive metabolism by CYP3A4 and CYP2D6. As a result, only 5% of ranolazine is eliminated unchanged. 73% percent of absorbed ranolazine is eliminated via kidneys, 25% is eliminated with feces [4,14]. The usual starting dose of ranolazine is 375 mg twice daily, which can be up-titrated to 750 mg twice daily. There are no precise dose recommendations for patients with mild and moderate renal dysfunction. However, it is noted that careful dose titration should be performed for the patients with CrCl 30–80 mL/min [4]. Ranolazine is a moderate to potent inhibitor of P-gp [11,12]. Until today, there was no data of interaction between dabigatran and ranolazine neither in literature, SmCP, or most of the drug interaction sites (for example: www.drugs.com) [15]. Even though on the Micromedex website, the interaction between dabigatran and ranolazine is described as possibly major, but available documentation is poor, only pharmacologic considerations lead clinicians to suspect the interaction exists [16]. In https://www.rxlist.com/drug-interaction-checker.htm it is reported that such interaction is possible and monitoring by a doctor is required [17]. Having renal impairment may cause an increase of ranolazine AUC up to 2 times [4,14], and when CrCl is lower than 30 mL/min ranolazine is contraindicated [4]. In this case, the patient with moderate renal impairment (CrCl 45.22 mL/min) was using the maximal recommended dose of ranolazine, and plasma ranolazine concentration may have been higher than usual and the effect on P-gp may have been strong. The plasma concentration of ranolazine, in this case, was not evaluated. This can be suspected by changes in the ECG. Ranolazine may prolong the QTc interval in a dose-dependent manner [4,13]. Mean changes in QTc after using 500 and 750 mg of ranolazine twice daily were 1.9 and 4.9 msec from baseline, respectively [11]. When the patient was administered to the hospital, the QTc interval was 473 msec. When renal function improved (CrCl increased from 45.22 mL/min to 58 mL/min), the QTc interval decreased to normal (414 msec). It is possible that ranolazine plasma concentration may have been highly increased on the hospitalization day and the effect on P-gp could be so strong that it caused a dabigatran overdose.

Based on Drugs Interaction Probability Scale (DIPS) [18,19], we evaluated this interaction 3 points out of 10, and this is a possible drug interaction.

## 4. Conclusions

The patient, who had renal failure and was using dabigatran together with ranolazine, had a significantly increased value of INR. It could be caused by the interaction between dabigatran and ranolazine, because dabigatran is a p-glycoprotein substrate, whereas ranolazine is the inhibitor of p-glycoprotein. These two medications should be used together with caution in patients with renal failure. It is recommended to use smaller doses of these drugs and to observe coagulation parameters if needed. In order to notice dabigatran overdosing sooner, thrombin time or aPPT should be evaluated, because the INR value significantly increases only when the plasma concentration of dabigatran is considerably higher than the therapeutic dose.

## Figures and Tables

**Table 1 medicina-56-00013-t001:** Change of laboratory results of the international normalized ratio (INR) and creatinine.

Date	INR	Creatinine (μmol/L)	CrCl Cockroft-Gault (mL/min)
Hospitalization day	7.03	168	45.22
1 day after	4.95	154	49.34
3 days after	2.88	131	58
6 days after	1.67	118	64.39
9 days after	1.33	146	52.04

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
