# Peer review of "Possible Interaction between Dabigatran and Ranolazine in Patients with Renal Failure"

_medicina, 2019, doi:10.3390/medicina56010013_

Round 1
Reviewer 1 Report
This a case report of the interplay between dabigatran, ranolazine and impaired renal function in a patient with heavy pathologies and medicines. The accumulation of dabigatran, indicated by the high INR level, and probably of ranozaline, as suggested by ECG, both drugs that share the same metabolic pathways, was not unexpected. Both were prescribed at full dose (150 mg bid for dabigatran, ranozaline 750 mg bid), whereas moderate renal impairment should have justified reduced dose. Microcytic anemia is suggestive of chronic GI bleeding, not unexpected with low dose aspirin on top of the anticoagulant treatment (association not recommended by the current guidelines for this patient). Taken together, the patient was at high risk of severe bleeding.
INR is not useful for monitoring direct oral anticoagulants, including dabigatran because it is poorly sensitive to this drug. The high INR level is suggestive of dabigatran levels >>> 1000 ng/mL, according to the INR reported in several cases of overdose in the literature. This is supported by the delay for recovering subnormal INR before discharge. In the absence of life-threatening bleeding, which would be an indication for idarucizumab, if available, time was the best antidote of dabigatran overdose, but haemodialysis was an option to shorten the period of high bleeding risk (not discussed in the paper).
This case report illustrates the risks of multiple treatments in such patients. Ranozaline joins the list of drugs that may interfere with oral direct anticoagulants in the pharmacovigilance registers.
The paper is well written and discussed. It could be interesting to describe the changes in treatment given to this patient at his discharge.
Author Response
Good day,
Thank You for the article review, comments and suggestions. I am sending You an updated case report.
The analysis of this cases was performed after discussion with hospital physician who treated this patient. Discussion was about the reason of increased INR in patient that was taking dabigatran. The patient has already been discharged from Hospital for a few weeks. The analysis was performed retrospectively using patient's medical records. Unfortunately, available patient's medical records did not contain information regarding the duration of exposure to dabigatran and ranolazine. It was unpossible to communicate with the patient. As result, the duration of exposure to both medications is unknown. Physician was recomended to use smaller doses of dabigatran and ranolazne, not to use aspirin together with dabigatran and perform blood aPPT test in order to observe possible dabigatran overdose.
Kind regards

Reviewer 2 Report
The authors reported the possible interaction between dabigatran and ranolazine in patient with renal failure. The topic is very interesting, and this case report provided useful information for clinical practice. Minor revision is needed before further consideration of publication.
1. How long did the patient take both dabigatran and ranolazine, respectively?
2. General information of ranolazine should be provided.
Author Response
Good day,
Thank you for the article review, comments and suggestions. I am sending You an updated case report.
The analysis of this case was performed after discussion with hospital physician who treated this patient. Discussion was about the reason of increased INR in patient that was taking dabigatran. The patient has already been discharged from Hospital for a few weeks. The analysis was performed retrospectively using patient’s medical records. Unfortunately, available patient’s medical records did not contain information regarding the duration of exposure to dabigatran and ranolazine. It was unpossible to communicate with the patient. As result, the duration of exposure to both medications is unknown. Physician was recomended to use smaller doses of dabigatran and ranolazne, not to use aspirin together with dabigatran and perform blood aPPT test in order to observe possible dabigatran overdose.
Kind regards
